# Enhanced Anti-Tumor Activity in Mice with Temozolomide-Resistant Human Glioblastoma Cell Line-Derived Xenograft Using SN-38-Incorporated Polymeric Microparticle

**DOI:** 10.3390/ijms22115557

**Published:** 2021-05-24

**Authors:** Tao-Chieh Yang, Shih-Jung Liu, Wei-Lun Lo, Shu-Mei Chen, Ya-Ling Tang, Yuan-Yun Tseng

**Affiliations:** 1Department of Neurosurgery, School of Medicine, Chung Shan Medical University Hospital, Taichung 40201, Taiwan; cshy1801@csh.org.tw; 2School of Medicine, Chung Shan Medical University, Taichung 40201, Taiwan; 3Department of Mechanical Engineering, Chang Gung University, Taoyuan 33302, Taiwan; shihjung@mail.cgu.edu.tw (S.-J.L.); vellick27candy@gmail.com (Y.-L.T.); 4Department of Orthopedic Surgery, Chang Gung Memorial Hospital-Linkou, Taoyuan 33302, Taiwan; 5Division of Neurosurgery, Department of Surgery, Shuang Ho Hospital, Taipei Medical University, New Taipei City 235041, Taiwan; 12317@s.tmu.edu.tw; 6Department of Surgery, School of Medicine, College of Medicine, Taipei Medical University, Taipei 110301, Taiwan; 190001@h.tmu.edu.tw; 7Department of Neurosurgery, Taipei Medical University Hospital, Taipei 110301, Taiwan

**Keywords:** glioblastoma multiforme (GBM), SN-38, interstitial chemotherapy, temozolomide-resistance, poly[(d,l)-lactide-co-glycolide] (PLGA)

## Abstract

Glioblastoma multiforme (GBM) has remained one of the most lethal and *challenging* cancers to treat. Previous studies have shown encouraging results when irinotecan was used in combination with temozolomide (TMZ) for treating GBM. However, irinotecan has a narrow therapeutic index: a slight dose increase in *irinotecan can induce toxicities that* outweigh its therapeutic benefits. SN-38 is the active metabolite of irinotecan that accounts for both its anti-tumor efficacy and toxicity. In our previous paper, we showed that SN-38 embedded into 50:50 biodegradable poly[(d,l)-lactide-co-glycolide] (PLGA) microparticles (SMPs) provides an efficient delivery and sustained release of SN-38 from SMPs in the brain tissues of rats. These properties of SMPs give them potential for therapeutic application due to their high efficacy and low toxicity. In this study, we tested the anti-tumor activity of SMP-based interstitial chemotherapy combined with TMZ using TMZ-resistant human glioblastoma cell line-derived xenograft models. Our data suggest that treatment in which SMPs are combined with TMZ reduces tumor growth and extends survival in mice bearing xenograft tumors derived from both TMZ-resistant and TMZ-sensitive human glioblastoma cell lines. Our findings demonstrate that combining SMPs with TMZ may have potential as a promising strategy for the treatment of GBM.

## 1. Introduction

Glioblastoma multiforme (GBM) is the most common type of malignant glioma (MG) in adults [1,2]. The median survival time for patients with GBM is only 12–15 months, with a low 5-year survival rate of less than 5% [2,3,4]. Surgery followed by radiotherapy with concomitantly administered chemotherapy is a mainstay in treatments for GBM. Temozolomide (TMZ) is generally used as a chemotherapeutic agent in clinical situations to treat patients with GBM [1,2]. However, TMZ resistance in GBM is one of the greatest challenges physicians encounter in treating GBM [1,2,5,6,7].

TMZ is an alkylating agent that shows anti-tumor activity across various cancers [5,6,7,8,9]. It has been demonstrated that TMZ exerts its anti-tumor activity on a variety of cell types, such as melanoma and glioma cells, by facilitating G2/M cell cycle arrest and cellular apoptosis through up-regulation of tumor suppression proteins [8,9]. In addition, TMZ is also found to generate O^6^*-*methylguanine (O6MeG), N^7^-methylguanine (N^7^-meG), and N^3^-methyladenine (N^3^-meA) in DNA. This eventually leads to DNA lesion-induced cytotoxicity, which results in cell death [2,8,10,11,12]. Evidence shows that DNA lesions caused by TMZ, such as O6MeG, activate the O^6^-methylguanine-DNA methyltransferase (MGMT)-mediated DNA repair process [8,9,11,12]. This, in turn, reduces the efficiency of TMZ in killing cancer cells. Indeed, a lack of MGMT gene promoter methylation, which leads to an elevated expression of functional MGMT proteins, has been observed in approximately half of GBM tumors and is linked to TMZ resistance in GBM [1,2,5,6,7,13]. Approved chemotherapeutic agents for treating GBM are very limited. Aside from TMZ, bis-chloroethylnitrosourea (BCNU, carmustine), is the only chemotherapeutic alkylating agent approved by the US Food and Drug Administration for treatment of GBM [10,13,14]. TMZ is administered orally, while BCNU is loaded into wafers of polifeprosan 20 (Gliadel, Guilford Pharmaceuticals-MGI Pharma, Bloomington, MN) and administered by being placed onto the brain cavity after removal of a brain tumor. BCNU has been considered an important treatment option for GBM since 1995, as its systemic toxicity was limited when Gliadel wafers were introduced to deliver BCNU [10,14]. While both of them are alkylating agents, BCNU is similar to TMZ in that it also induces DNA damage through alkylating DNA at the O^6^ position of guanine [8,10,11,12,13]. Therefore, it is not surprising that the therapeutic effectiveness of BCNU is greatly attenuated in patients with high MGMT expression [10]. Indeed, several studies on TMZ have shown that TMZ-resistant phenotypes also remain less sensitive to BCNU [15,16]. These results indicate that drugs with similar mechanisms of action become less effective for treating GBM patients who do not respond to TMZ treatment. Thus, more effective anticancer therapeutic options with different mechanisms of action are needed for treating GBM.

Topoisomerase I (TOP I) inhibitors merit further consideration as a potential alternative therapeutic option for GBM patients with regard to generating anti-tumor activity via different mechanisms and pathways [17,18,19]. TOP I is a ubiquitous enzyme that relaxes both positive and negative DNA supercoiling [20]. Its function is therefore essentially required during DNA transcription, replication, and repair [20]. After the DNA is relaxed, TOP I spontaneously religates the DNA breaks [20]. This TOP I-mediated religation process can be inhibited by TOP I inhibitors, such as irinotecan [1], which prevent religation of the DNA breaks and lead to cellular apoptosis [20]. In fact, this ability of irinotecan to induce apoptosis has been utilized in developing it as an anticancer drug across various cancers, including colorectal and ovarian cancers [21,22]. With regard to recurrent glioma, about 16% and 21% of patients, respectively, respond to treatment of irinotecan in combination with other drugs, such as celecoxib and BCNU, while the vast majority of patients do not [1,23]. Irinotecan and TMZ combination therapy has also been tested in a phase II trial for GBM, with an objective response rate of 19% [24]. Overall, these results are encouraging, but less than what was hoped for when we take into consideration all aspects of combining two drugs that act by distinct mechanisms of action. Part of the reason for these results may be due to the narrow therapeutic index of irinotecan and the low conversion rate of irinotecan to SN-38 (7-ethyl-10-hydroxycamptothecin) in patients [25,26,27]. Irinotecan is a prodrug, and it has been shown that it can be metabolized into SN-38 by carboxyesterases in colorectal cancer and glioma cells [22,25,27]. SN-38 is the active metabolite of irinotecan that accounts for both its anti-tumor efficacy and toxicities [27,28]. The high toxicity and instability of SN-38 in the physiological environment (pH 7.4) appear to be the major limiting factors for successful treatment of GBM [28]. It has been suggested that drug carriers that offer a slow but sustained release of drugs may overcome the toxicities and instability of SN-38 [28,29,30]. However, minimal effort has been made to customize the local delivery of this hydrophobic anticancer drug to the brain.

In a previous study, we loaded SN-38 into poly(lactid-co-glycolid) (PLGA) microparticles (SMPs) using an electrospraying technique. SMPs were introduced into rat brain parenchyma via stereotactic techniques [31]. Our previous results suggested that SMPs are biocompatible and biodegradable [31]. SMPs were delivered intratumorally into tumor-bearing rats via stereotactical techniques; the results showed a sustained drug release pattern and significant therapeutic efficacy [31]. Most importantly, this drug delivery system has minimal systemic side effects [31]. Given the low toxicity of SN-38, using SMP-based interstitial delivery with SN-38 functions through a different mechanism of action than TMZ. Therefore, we sought to determine whether higher treatment efficacy could be achieved when SN-38 and TMZ were used in combination to treat TMZ-resistant glioblastomas. We evaluated this therapy and compared it with other treatment regimens in xenografts that are either derived from TMZ-resistant (U87TR) or TMZ-sensitive (U87) human glioblastoma cell lines. Our results suggest that administering SN-38 with an SMP-based interstitial delivery system combined with oral TMZ significantly reduces tumor growth and extends survival in mice with either TMZ-resistant or TMZ-sensitive glioblastomas as compared to other treatment regimens, including Gliadel wafer combined with oral TMZ. We verified that switching drug treatment to SN-38, which has a different mechanism of action than TMZ, has the potential to improve the outcome for patients with GBM who are considered resistant to TMZ treatment.

## 2. Results

### 2.1. Characteristics of SN-38-Incorporated Microparticles

The biodegradable SN-38-embedded PLGA microparticles were successfully fabricated through electrospraying techniques. The scanning electron microscopy (SEM) image of SMPs shows that SMPs possess an average diameter of 1.26 ± 0.78 µm (range: 760 nm to 6.43 µm) (Figure 1A). The zeta potential and polydispersity index were −0.84 ± 0.1 mV and 3.657, respectively. Injection procedures and materials (SMPs) induced a transient inflammation reaction, i.e., infiltrated leukocytes surrounded the SMPs and injection site. These infiltrated leukocytes were mainly present within the first 4 weeks but resolved after 8 weeks (Figure 1B). The in vivo study demonstrated that SMPs could release high concentrations of SN-38 in brain tissue (local) and much lower SN-38 concentrations in blood (systemic) over 8 weeks (Figure 1C). More detailed information was presented in our previous study [32].

### 2.2. Cytotoxicity of SMPs in Human U87 and Astrocytes-Hippocampal (HA-h) Cell Lines

To assess the cytotoxic effect of SMPs, human U87, U87-TR and astrocytes-hippocampal (HA-h) cells were treated with SMPs at doses ranging from 0.1 nM to 1000 nM for 48 and 72 h (Figure 1D). The inhibitory effect of SMPs on the cell viability of these three cell lines was examined using an MTT assay (Figure 1D). A decrease in cell viability was observed in all three cell lines after 24 and 48 h exposure to increasing concentrations of SMPs (0.1–1000 nM). However, the inhibitory effect of SMPs on U87 cell or U87-TR viability was more pronounced as compared to HA-h cells. As shown in Figure 1D, the 50% inhibitory concentration (IC50) values of SMPs for U87 cells were 4.06 nM (48 h) and 0.38 nM (72 h), while IC50 of SMPs for U87-TR cells were 669.46 nM (48 h) and 58.86 nM (72 h). The results suggested that SMPs inhibit the growth of U87, U87-TR and HA-h cells in a dose- and time-dependent manner.

### 2.3. Combination Treatment of TMZ and SMPs Prolongs the Survival of *Mice Bearing* U87 or U87-TR Xenografts

To evaluate the combination effect of TMZ and SMPs in comparison with TMZ alone or TMZ and Gliadel, mice bearing U87 xenografts (TMZ-sensitive) were treated with vehicle, Gliadel, or SMPs (n = 8 per each group) in combination with orally administered TMZ after completing drug treatment in cycle 1, where we pretreated the tumor-bearing mice with orally administered TMZ. For the vehicle in combination with orally administered TMZ (TMZ vehicle) treatment cohort, mice bearing U87 xenografts exhibited the lowest median survival (Figure 2A). They survived for 41.75 ± 11.68 days while mice in the TMZ-combined Gliadel and TMZ-combined SMP treatment cohorts survived 47.00 ± 9.17 days and 63.17 ± 8.74 days, respectively (TMZ vehicle versus TMZ-combined Gliadel, *p* = 0.038; TMZ vehicle versus TMZ-combined SMP, *p* = 0.0098; TMZ-combined Gliadel versus TMZ-combined SMP, *p* = 0.028). In a similar fashion, TMZ-combined SMP treatment continued to show a survival benefit in mice bearing U87–TR xenografts (TMZ-resistant) as compared to the other two treatment cohorts, i.e., the TMZ vehicle and TMZ-combined Gliadel treatment cohorts. The TMZ-combined SMP treatment cohort showed a statistically significant benefit compared with the other two treatment cohorts, with a median survival of 39.79 ± 8.63 days compared to the 27.14 ± 6.67 days for the TMZ vehicle treatment cohort and 31.38 ± 5.59 days for the TMZ-combined Gliadel treatment cohort (Figure 2B; TMZ vehicle versus TMZ-combined Gliadel, *p* = 0.142; TMZ vehicle versus TMZ-combined SMP, *p* = 0.0371; TMZ-combined Gliadel versus TMZ-combined SMP, *p* = 0.0465.). Taken together, these findings suggest that the combination therapy of TMZ and SMPs improves survival rates in both TMZ-sensitive and TMZ-resistant xenograft models compared to the TMZ vehicle and TMZ-combined Gliadel treatment cohorts.

### 2.4. Combination Treatment of TMZ and SMPs Suppresses Tumor Growth in both TMZ-Sensitive and TMZ-Resistant Xenograft Models

Luciferase-positive TMZ-sensitive and TMZ-resistant xenografted tumors enabled the detection of tumors in the brains of nude mice by serial bioluminescence imaging. The kinetics of tumor growth and response to therapy are shown in Figure 3A,B. In both TMZ-sensitive (Figure 3A and Figure 4) and TMZ-resistant (Figure 3B and Figure 5) xenograft models, mice from the TMZ vehicle and TMZ-combined Gliadel treatment cohorts showed a relatively rapid increase in bioluminescent signals in their tumors over time (Figure 3A,B, Figure 4 and Figure 5). By contrast, the TMZ-combined SMP treatment cohort exhibited a substantial reduction in tumor growth that was observed in both U87 and U87-TR xenograft models (Figure 3A,B, Figure 4 and Figure 5). Our data suggested that SMPs combined with orally administered TMZ could slow tumor growth in mice bearing U87 or U87-TR xenografts.

According to the IVIS images, spinal metastases were detected in U87-TR xenograft mice. Four nude mice bearing U87-TR xenografts that received TMZ vehicle treatment and one nude mouse bearing U87-TR xenografts that received TMZ-combined Gliadel treatment were found with spinal metastases. No spinal metastases were noticed in nude mice bearing U87-TR xenografts that received TMZ-combined SMP treatment (*p* < 0.05) (Figure 5). Overall, no spinal metastases were detected in mice bearing U87 xenografts regardless of which treatment the mice received (Figure 4).

### 2.5. The Presence of a Central Necrosis Area and GFAP Expression Changes in both TMZ-Sensitive and TMZ-Resistant Xenografted Tumors after a Combination Treatment of TMZ and SMPs

After determining the bioluminescent intensity of each tumor and the mean intensity value of all the tumors, we used the mice (one per treatment cohort) whose tumors were closest to that mean value within their respective treatment cohorts to assess the histological changes in tumors due to different treatments. The brain tissues of these mice were meticulously and surgically extracted at 2 weeks after treatment. The histological analysis of tumors revealed the presence of central (coagulative) necrosis areas, which were more extensively found in mice bearing U87 or U87-TR xenografts that had the TMZ vehicle treatment (Figure 6A,D). In the TMZ-combined Gliadel treatment cohort, a restricted tumor area and small central necrosis were observed in both xenograft models (Figure 6B,E). Tumors were only found in small areas within the brains of both xenograft models when mice were treated with a combination of TMZ and SMPs (Figure 6C,F). The changes in central (coagulative) necrosis areas in these tumors were due to different treatments. They were further characterized by performing *i*mmunohistochemistry (IHC) staining using an antibody raised against glial fibrillary acidic protein expression (GFAP), where expression is associated with a less aggressive tumor. In the TMZ vehicle and TMZ-combined Gliadel treatment cohorts, weak and no expression of this protein were found in TMZ-sensitive and resistant xenografted tumors, respectively (Figure 6G,H,J,K). Coarse, dendritic GFAP-positive glial cells were noted in both TMZ-sensitive and TMZ-resistant xenografted tumors from the TMZ-combined SMP treatment cohort (Figure 6I,L).

### 2.6. Combination Treatment of TMZ and SMPs Inhibited Cell Proliferation and Induced Apoptosis in both TMZ-Sensitive and TMZ-Resistant Xenograft Models

Ki-67 expression was used to evaluate the proliferation of tumor cells. An extremely high Ki-67 labeling index was noted in the TMZ vehicle and TMZ-combined Gliadel treatment cohorts for both TMZ-sensitive and TMZ-resistant xenograft models (Figure 7A,B,D,E). The expression of Ki-67 in both xenograft models treated with a combination of TMZ and SMPs was significantly lower among all treatment cohorts (Figure 7C,F). A TUNEL assay was performed to evaluate the combination effect of TMZ and SMPs on tumor cell apoptosis. In both TMZ-sensitive and TMZ-resistant xenograft models, scarce apoptotic nuclei were found in the TMZ vehicle treatment cohort (Figure 7G,J), whereas there were some apoptotic nuclei found within the intratumoral areas in the TMZ-combined Gliadel treatment cohort (Figure 7H,K). A significant increase in TUNEL-positive apoptotic nuclei was observed in both xenograft models treated with a combination of TMZ and SMPs (Figure 7I,L). The results demonstrated that the combination treatment of TMZ and SMPs sensitizes glioma tumor cells to apoptosis and decreases proliferation.

## 3. Discussion

The chemotherapeutic options for patients with GBM have almost exclusively relied on TMZ and BCNU [10,14]. Based on preclinical data in a phase I study, a synergistic anti-tumor effect has been demonstrated for TMZ and BCNU in patients with glioma [33,34]. Nevertheless, there are many studies showing similar results in which a combination of TMZ and BCNU is a promising therapeutic strategy to improve anti-tumor efficacy in glioma [33,34], and our findings are in close agreement with these studies when TMZ-combined Gliadel treatment was used to treat mice bearing TMZ-sensitive xenografts as compared to TMZ vehicle treatment (Figure 2A and Figure 3A). However, Gliadel combined with TMZ neither demonstrated a survival benefit (Figure 2B) nor an anti-tumor activity (Figure 3B) when comparing TMZ-combined Gliadel treatment to TMZ vehicle treatment in a TMZ-resistant xenograft model. Indeed, several prospective and retrospective studies also suggest that a combination of TMZ and BCNU has *no* beneficial effect in patients with high-grade glioma [35,36]. These findings, therefore, suggest that the combination of TMZ and Gliadel offers little or no benefit in TMZ-resistant xenograft models. This may be attributed to both of these drugs being alkylating agents, which cause cell death in a similar fashion. For this reason, there is an urgent demand for using a combination of drugs that exert anti-tumor activities through targeting different pathways.

TOP I is an important therapeutic target in cancer [20]. Top I inhibitors, such as SN-38, induce cellular apoptosis through a mechanism of action that is distinct from TMZ and Gliadel [8,10,11,12,13,20]. However, the high toxicity and instability of SN-38 in the physiological environment (pH 7.4) have made its clinical use infeasible [28]. In our previous study, we demonstrated that SMP-based interstitial delivery of SN-38 is feasible [31]. This therapeutic strategy, combined with orally administered TMZ, displayed increased anti-tumor activity in a rat model of malignant glioma. Since it has remained unclear whether SMPs combined with orally administered TMZ will be useful for overcoming temozolomide resistance in glioblastoma, we addressed this question using SMP-based interstitial delivery of SN-38 in xenografts derived from TMZ-resistant human glioblastoma cell lines. Our results indicate that this combination therapy provides therapeutic anti-tumor benefits not only for TMZ-sensitive xenografts, but also for TMZ-resistant xenografts. Both xenograft models showed significant reduction in tumor burden (Figure 3A,B, Figure 4 and Figure 5) that corresponded with significant improvement of animal survival (Figure 2) compared with the TMZ vehicle and TMZ-combined Gliadel treatment cohorts.

TMZ and SN-38 are known to inhibit tumor cell proliferation and induce apoptosis across various types of cancers [37,38]. In our study, suppression of tumor growth through TMZ-combined SMP treatment (Figure 3A,B, Figure 4, and Figure 5) corresponded to the lowest amount of Ki67 positive cells (Figure 7C,F). Meanwhile, the highest amount of TUNEL-positive cells was comparable to the TMZ vehicle and TMZ-combined Gliadel treatment cohorts in both TMZ-sensitive and TMZ-resistant xenograft models (Figure 7G,H,J,K), as analyzed with immunohistochemistry. In addition, TMZ-combined SMP treatment reduces the area of coagulative necrosis (Figure 6C,F). Necrosis is associated with poor prognosis in patients with glioblastoma, and the increase in the area of coagulative necrosis in tumors could be due to abnormal cell survival and resistance to apoptosis [33,39,40,41,42]. Hence, our data may suggest that the reduction in the area of coagulative necrosis found in both xenografted tumors is due to the increased apoptosis caused by TMZ-combined SMP treatment. GFAP is the principal indicator of astroglial cells; the number of cells expressing GFAP is inversely proportional to the extent of anaplasia in astrocytic neoplasms [42,43]. The loss of GFAP expression has been proposed to be highly relevant to glioma development and progression [42,43]. Increased expression of GFAP in TMZ-sensitive (Figure 6I) and TMZ-resistant (Figure 6L) xenograft tumors after TMZ-combined SMP treatment suggests that this drug combination may affect the processes of cell differentiation and reduce the malignancy of human glioma cells.

The dissemination of MG to the spinal compartment, although uncommon, has an even poorer outcome [32]. When MG is under adequate treatment and control, metastases of intracranial MG to the spinal cord are detected relatively rarely [32,44]. Spinal metastases were found in four nude mice bearing U87-TR xenografts that received TMZ vehicle treatment and one nude mouse bearing U87-TR xenografts that received TMZ-combined Gliadel treatment (*p* < 0.05) (Figure 5). Overall, regardless of which treatment the mice received, no spinal metastases were detected in mice bearing U87 xenografts (Figure 4). U87-TR glioma cells are more infiltrative and malignant than U87 glioma cells. However, we did not observe any spinal metastases in mice bearing U87-TR xenografts that received TMZ-combined SMP treatment (Figure 5). These results demonstrate that the control of tumor growth and progression in mice bearing U87TR xenografts was more effective with the intratumoral TMZ-combined SMP treatment.

The present study offers experimental validation of the feasibility of using SMP-based interstitial delivery of SN-38 to reduce the growth of xenografted tumors that are derived from TMZ-sensitive and TMZ-resistant human glioblastoma cell lines. Similar to the administration of Gliadel wafers, SN-38 can be introduced to the targeted site (tumor) via stereotactical techniques. By adopting these stereotactical techniques for the delivery of SN-38, SN-38 can be released to tumor tissues in an effective concentration for the appropriate duration of time; this ensures high anticancer activity and low systemic side effects. Furthermore, stereotactic injection can be repeated safely through a small burr hole surgery and is considered the procedure of choice in cases of multiple tumors and/or tumors located deep in the brains. In such cases, an open surgical approach is impossible and/or perilous, especially when the tumor is located in crucial areas of the brain. Thus, with regard to ease of administration, SMP could be a drug of choice for GBM treatment that is comparable to Gliadel wafers. We demonstrated that TMZ-combined SMP treatment has the most beneficial anti-tumor effect when compared to the other two treatments, suggesting that SMPs could be a better choice of drug for GBM treatment than Gliadel wafers. SMPs have a number of favorable features in combination with TMZ, but the most important one is that SMPs possess mechanisms of tumor-killing action distinct with TMZ. This may overcome drug resistance and improve the efficacy of using two alkylating agents, such as Gliadel wafers combined with TMZ.

## 4. Materials and Methods

### 4.1. Drugs and Chemicals

Poly(lactid-*co*-glycolid) polymers with a lactide:glycolide ratio of 50:50 (Resomer^®^ RG503) were purchased from Boehringer Ingelheim (Ingelheim am Rhein, Germany). An amount of 20 mg of Temozolomide (TMZ) was commercially acquired from Rising Pharm. Inc. (East Brunswick, NJ, USA.). Gliadel wafer was purchased from MGI Pharma, Inc. (Bloomington, IN, USA), and SN-38 was purchased from Sigma–Aldrich (Saint Louis, MO, USA). *A pathology* assessment of tumor tissues, including terminal deoxynucleotidyl transferase dUTP nick end labeling (TUNEL) assay, hematoxylin and eosin (H&E) staining, glial fibrillary acidic protein expression (GFAP), and Ki-67 was carried out by a professional analysis service (Biotools Co., New Taipei, Taiwan).

### 4.2. Human *Glioma* Cell Lines and Cell Cultures

Luciferase-expressed U87 and U87 temozolomide-resistant (U87-TR) cells were provided by Dr. Kwang-Yu Change (National Health Research Institute, Tainan, Taiwan), and U87-TR cells were established as described previously [5,20,45]. The astrocytes-hippocampal (HA-h), U87, and U87-TR cells were maintained in DMEM containing 10% FBS, 100 µg/mL penicillin G, and 100 streptomycin. 

### 4.3. Experimental Animals

All procedures related to animals were approved by the Institutional Animal Care and Use Committee of Taipei Medical University (LAC-2019-0523, approved date: Dec. 23, 2019). All animal experiments were performed in accordance with the guidelines of the Department of Health and Welfare, Taiwan. Efforts were made to minimize the number of animals used and their suffering. A total of 60 male NOD-SCID male mice (8 weeks old) were commercially acquired from BioLASCO Taiwan Co., Ltd. (Taipei, Taiwan).

### 4.4. Preparation of SN-38 Incorporated Polymeric Microparticles (SMPs)

The detailed preparation of both virgin PLGA microparticles and SMPs and the process of incorporating SN-38 into the PLGA microparticles were described in our previous publication [31]. In brief, the electrosprayed virgin PLGA microparticles were prepared by dissolving 500 mg of PLGA in 10 mL of dichloromethane. Fabrication of SMPs was prepared by dissolving 300 mg of PLGA and 50 mg of SN-38 in 1 mL of dichloromethane for 1 h. The mixed solution was then transported with a pump with a volumetric flow rate of 0.9 mL/h. The needle was connected to a high voltage supply with a positive direct current voltage of 8 kV. The distance between the needle tip and collected ground electrode was 13 cm. All electrospraying experiments were completed under a temperature of 25 °C and a relative humidity of 60%.

### 4.5. In Vivo Elution Characteristics of SMPs

The 2.4 mg of SMPs were mixed with 10 µL of DMSO, and the mixture was stereotactically injected slowly into the cerebral parenchyma by using a syringe infusion pump after performing a small burr hole. Forty rats were randomly divided into nine groups of three or four rats, with each group being euthanized after a certain period (3 days and 1, 2, 3, 4, 5, 6, 7, and 8 weeks). Approximately 0.5 mL blood specimens through cardiac puncture were collected. Ipsilateral injected with SMNs was surgically extracted. Approximately 50 mg of brain parenchymal specimen was obtained by carefully extracting the brain. The tissue specimens were extracted through sonication for 20 s and then centrifuged. Plasma was collected and maintained at −80 °C. The concentration of SN-38 in the collected solutions was evaluated using a Hitachi L-2200 HPLC assay (Multisolvent Delivery System, Tokyo, Japan). More detailed methods and in vitro elution characteristics of SN-38 were described in our previous publication [31].

### 4.6. Cell Viability Assay

An MTS Assay Kit (Abcam, ab197010) was used for measurements of cell viability. Cells were seeded into 96-well plates at 5000 cells/well and incubated overnight. After treatment with SMPs for 48 and 72 h, MTS reagent was added to the cell culture media and incubated for 1.5 h at 37 °C. Finally, the optical density (OD) of the solution was determined with a spectrometer at a wavelength of 490 nm. All frig-treated cells were normalized with their respective untreated control cells, which were considered as 100% viable. Data were expressed as mean ± SD of triplicate experiments. IC50 values were calculated with GraphPad Prism v8.3.0 (GraphPad Inc., La Jolla, CA, USA). 

### 4.7. Generation of Human *Glioma* Cell-Derived TMZ-Sensitive and TMZ-Resistant Mouse Xenografts

Experimental procedures related to the intracranial transplantation of GBM cells were taken from a previous study (Hevener et al., 2018). All NOD-SCID mice were anesthetized through halothane inhalation. Approximately 6 mm length linear scalp incision was performed lateral to the midline. The undermined muscle and scalp fascia were dissected, and a burr hole (approximately 1.0 mm in diameter) was drilled using an electric burr. The mice were randomly divided into U87 (TMZ-sensitive) and U87-TR (TMZ-resistant) groups, and the luciferase-expressed U87 cells (2 × 10^5^) and U87-TR cells (2 × 10^5^) were respectively injected into the brain cortex at a depth of 3 mm using stereotactic instruments (Model: 68001, RWD Life Science Inc., San Diego, CA, USA) and a 10-µL sterile Hamilton Neuro syringe (Hamilton, Reno, NV, USA). Animal weight and behavior were monitored every 2 days. The luciferase activity of U87 and U87-TR were monitored by the IVIS 200 system (Xenogen Corporation, Alameda, CA, USA). Three mice that died within 48 h after the operation (perioperative period) were excluded from the study, and *no* intracranial hemorrhage or abscess formation-related deaths were observed in mice bearing U87 or U87-TR xenografts. Of the successfully established xenografted tumors, there were 28 in mice bearing U87 xenografts and 27 in mice bearing U87-TR xenografts. We utilized 24 mice from each group for survival and tumor growth rate analyses. Four mice in U87 xenografts and three mice in U87-TR xenografts were used for assessing the histopathology and morphologic characteristics of the tumor tissues.

### 4.8. Animal Grouping and Drug Treatment Protocols

Ten days after the injection of luciferase-expressed U87 or U87-TR cells in NOD-SCID male mice, IVIS examination was performed to confirm that the mice bearing U87 (TMZ-sensitive) or U87-TR (TMZ-resistant) xenografts were successfully established. Thereafter, all mice bearing U87 or U87-TR xenografts received oral TMZ therapy throughout all treatment cycles (i.e., cycle 1 and cycle 2). TMZ (Temodal 100 mg) was orally administered to all mice bearing U87 or U87-TR xenografts at a dosage of 200 mg/BSA (m^2^) once daily for the first 5 days, followed by a 23-day treatment interruption (cycle 1); if the mice bearing U87 or U87-TR xenografts survived longer than 1 month, cycle 2 was initiated. The mice bearing U87 or U87-TR xenografts were randomly assigned to one of three experimental groups to receive different treatments in combination with orally administered TMZ. The groups were set up as follows. (1) The TMZ vehicle treatment cohort: mice in this control group were given an injection of vehicle (3 µL of dimethyl sulfoxide (DMSO) mixed with 0.6 mg pure PLGA microparticle) through the previous burr hole. (2) The TMZ-combined Gliadel treatment cohort: mice in this group were given a Gliadel treatment using a stereotactic intratumoral injection of 3 µL of DMSO mixed with one-eightieth Gliadel (2.5 mg) that contains 96.25 µg of BCNU. (3) The TMZ-combined SMP treatment cohort: mice in this group were given SMP treatment using stereotactic intratumoral injection of 3 µL of DMSO mixed with SMPs—i.e., 0.6 mg of PLGA microparticles containing 100 µg of SN-38. IVIS examinations were performed to monitor the luciferase activity of xenografted tumors at 2, 5, 9, 12, 16, 19, 23, 26, 30, 37, and 44 days after treatment.

### 4.9. *Pathology* Assessment of Tumor Tissues

The brain parenchyma samples were meticulously extracted, fixed with 10% formalin, and enclosed in paraffin. Subsequently, the brain tissue specimens were immersed in 10% buffered formal saline before routine embedding in paraffin and microscopic evaluation of H&E-stained 5-µm-thick sections. The Ki-67 labeling index, evaluated through MIB-1 immunostaining, was expressed as a percentage derived by counting the number of positively stained nuclei in 1000 tumor cells pooled from 3–5 fields (each having an area of 0.16 mm^2^) and examined at high-power magnification. Immunocytochemical staining of antibodies raised against GFAP (glial fibrillary acidic protein) expression was also performed on these sections. In situ TUNEL assay was used to evaluate apoptosis in tissue sections. Pathological studies, including hematoxylin and eosin (H&E) staining, GFAP, Ki-67 labeling index, and TUNEL assay, were carried out by a professional analysis agent (Biotools Co., New Taipei, Taiwan).

### 4.10. Statistical Analysis

Statistical analyses were performed using GraphPad Prism v8.3.0 (GraphPad Inc., La Jolla, CA, USA). Data were expressed as mean ± standard deviation unless otherwise specified. A paired-sample *t*-test was used to identify statistically significant differences between groups. The statistical significance of Kaplan–Meier survival curves was determined using a post-hoc log-rank test. A two-way ANOVA was adopted to compare luciferase activity in three groups at multiple time points. For all the comparisons in this study, a *p* value of less than 0.05 was considered statistically significant. 

## 5. Conclusions

GBM presents a major therapeutic challenge because of its poorly circumscribed ability to infiltrate healthy brain parenchyma, and it is refractory to surgery, radiosurgery, and conventional chemotherapy. In fact, TMZ resistance has been observed in approximately half of GBM patients. Thus, innovative therapeutic agents and strategies are imperative for improving therapeutic efficacy and overcoming TMZ resistance. In this study, we validated the feasibility of using SMP-based interstitial delivery of SN-38 to reduce the growth of xenografted tumors derived from TMZ-sensitive and TMZ-resistant human glioblastoma cell lines. Our data revealed that TMZ-combined SMP treatment renders a survival benefit in both TMZ-sensitive and TMZ-resistant xenograft models. In addition, TMZ-combined SMP treatment also significantly reduces tumor growth rates, decreases the area of tumor necrosis, and increases apoptosis compared to the other two treatment cohorts, i.e., the TMZ vehicle and TMZ-combined Gliadel treatment cohorts. Taken together, our findings suggest that in combination with SMPs, TMZ may have potential as a promising alternative strategy for the treatment of GBM, especially for TMZ-resistant GBM.

## Figures and Tables

**Figure 1 ijms-22-05557-f001:**
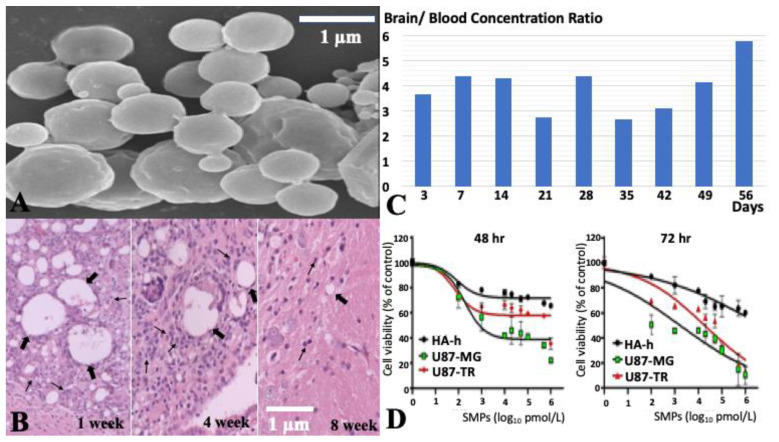
Characterization of SMPs. (**A**) Scanning electron microscopy (SEM) image of SMPs. The average macroparticle size is about 1.26 ± 0.78 µm in diameter. (**B**) H&E staining of brain region after injection of SMPs. Injection of SMPs induced transient leukocytes accumulation at the site of injection and around SMPs within the first 4 weeks but resolved after 8 weeks. The thick arrows indicated the injected SMPs and thin arrows indicated leukocytes. (**C**) In vivo release study. In vivo release of SN-38 from biodegradable SMPs; the brain/blood concentration ratio was between 2.65–5.76. (**D**) MTT assay. The graph represents the cytotoxic effect of SMPs in HA-h, U87-TR and U87-MG cell lines at different concentrations (0–6 log_10_ pmol/L) after 48 and 72 h incubation. Results are expressed as a percentage of control ± SEM from three independent experiments.

**Figure 2 ijms-22-05557-f002:**
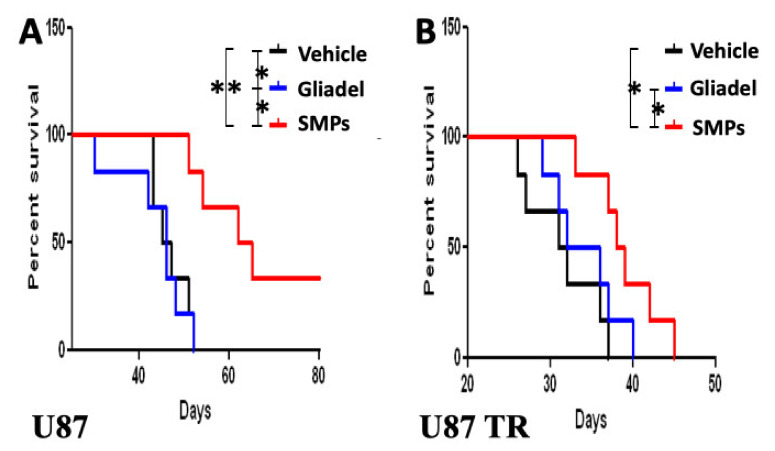
Kaplan–Meier survival curves of tumor-bearing mice following treatment with TMZ vehicle, TMZ-combined Gliadel, and TMZ-combined SMPs. (**A**) Survival curves of mice bearing U87 xenografts. (**B**) Survival curves of mice bearing U87-TR xenografts. The study was powered with eight mice per treatment arm for statistical significance. Log-rank (Mantel–Cox) tests were performed on survival plots. A Student t-test was performed to quantify differences in treatment arms (* *p*  ≤  0.05; ** *p*  ≤  0.01).

**Figure 3 ijms-22-05557-f003:**
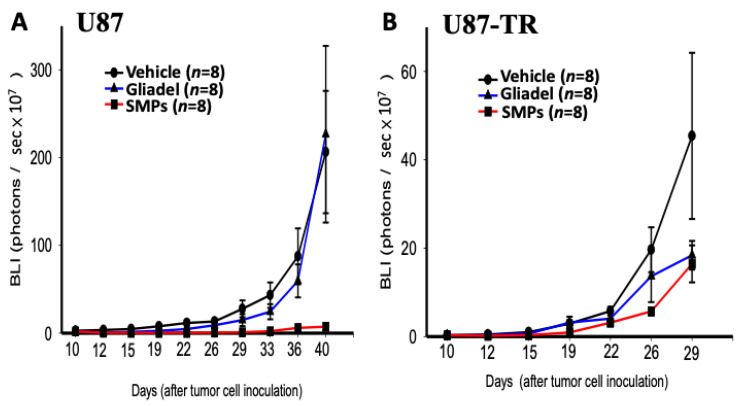
Quantification of average tumor bioluminescence values throughout the course of treatment with TMZ vehicle, TMZ-combined Gliadel, and TMZ-combined SMPs. (**A**) Mice bearing U87 xenografts. The bioluminescent intensity of tumors was significantly lower in the mice treated with TMZ-combined SMP treatment since day 12 compared to TMZ vehicle treatment (*p* = 0.0015); TMZ-combined SMP treatment since day 19 compared to TMZ-combined Gliadel treatment (*p* = 0.038); TMZ-combined Gliadel treatment since day 19 compared to TMZ vehicle treatment (*p* = 0.0145). Comparison of the inhibition of tumor growth following different treatments during the period of 10–40 days, the bioluminescent intensity of tumors was significantly lower in the mice treated with TMZ-combined SMP treatment compared to TMZ vehicle treatment (*p* < 0.001); TMZ-combined SMP treatment compared to TMZ-combined Gliadel treatment (*p* = 0.0017); TMZ-combined Gliadel treatment compared to TMZ vehicle treatment (*p* = 0.0143). (**B**) U87-TR xenografts. The bioluminescent intensity of tumors was significantly lower in the mice treated with TMZ-combined SMP treatment since day 12 compared to TMZ vehicle treatment (*p* = 0.0015); TMZ-combined SMP treatment since day 19 compared to TMZ-combined Gliadel treatment (*p* = 0.0183); TMZ-combined Gliadel treatment since day 12 compared to TMZ vehicle treatment (*p* = 0.0362). Comparison of the inhibition of tumor growth following different treatments during the period of 10–26 days, the bioluminescent intensity of tumors was significantly lower in the mice treated with TMZ-combined SMP treatment compared to TMZ vehicle treatment (*p* < 0.001); TMZ-combined SMP treatment compared to TMZ-combined Gliadel treatment (*p* = 0.00363); TMZ-combined Gliadel treatment compared to TMZ vehicle treatment (*p* < 0.001). The study was powered with eight mice per treatment cohort for statistical significance. Log-rank (Mantel–Cox) tests were performed on survival plots. A Student *t*-test was used to quantify differences in treatment arms.

**Figure 4 ijms-22-05557-f004:**
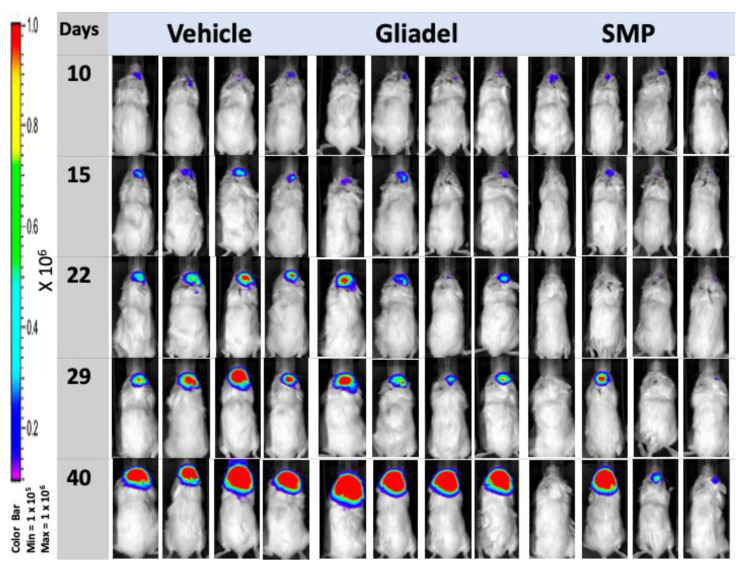
IVIS images of mice bearing U87 xenografts. Representative bioluminescent images of mice bearing U87 xenografts were taken on days 10, 15, 22, 29, and 40 following treatment with TMZ vehicle, TMZ-combined Gliadel, or TMZ-combined SMPs.

**Figure 5 ijms-22-05557-f005:**
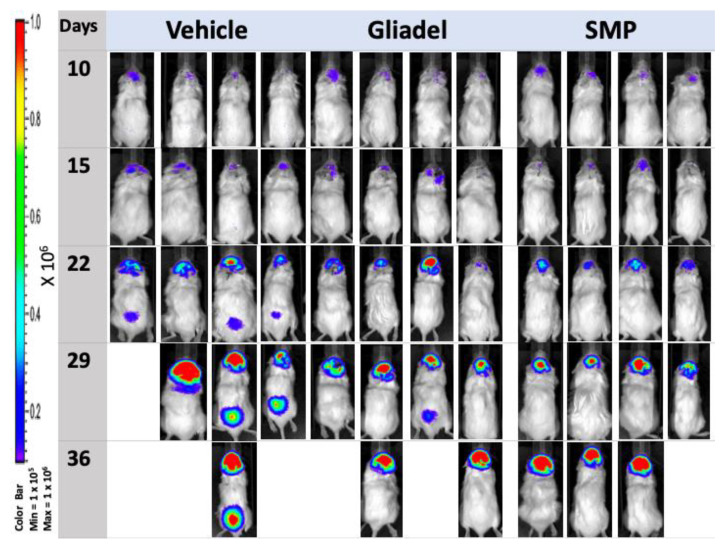
IVIS images of mice bearing U87-TR xenografts. Representative bioluminescent images of mice bearing U87-TR xenografts were taken on days 10, 15, 22, 29, and 36 following treatment with TMZ vehicle, TMZ-combined Gliadel, or TMZ-combined SMPs.

**Figure 6 ijms-22-05557-f006:**
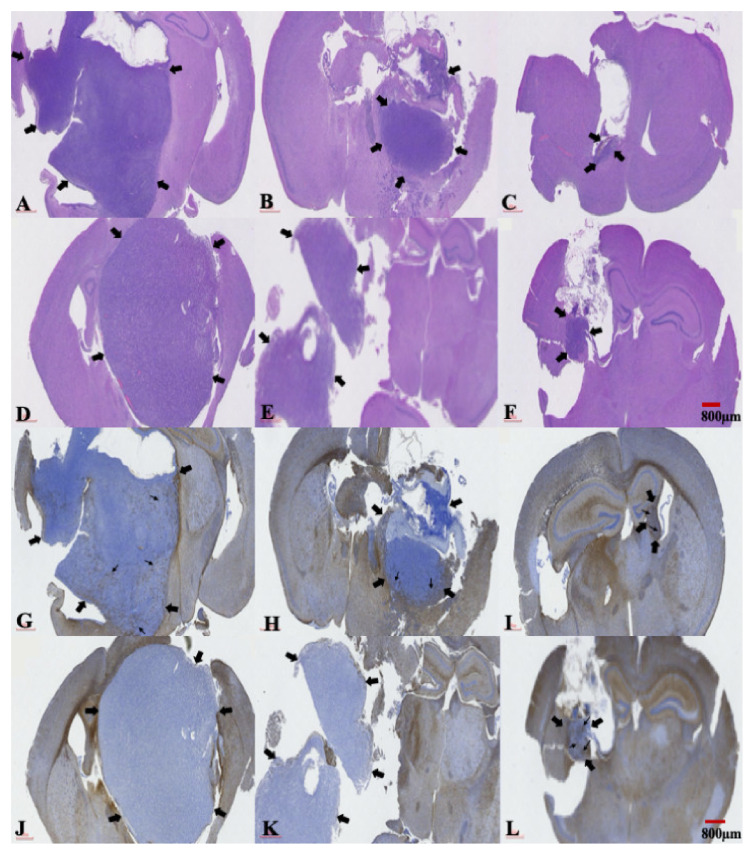
Staining of U87 and U87-TR xenografted tumor regions following treatment with TMZ vehicle, TMZ-combined Gliadel, or TMZ-combined SMPs. (**A**–**F**) represent H&E staining and (**G**–**L**) represent GFAP immunocytochemical staining. Mice bearing U87 xenografts treated with TMZ vehicle: (**A**,**G**); TMZ-combined Gliadel: (**B**,**H**); TMZ-combined SMPs: (**C**,**I**). Mice bearing U87-TR xenografts treated with TMZ vehicle: (**D**,**J**); TMZ-combined Gliadel: (**E**,**K**); TMZ-combined SMPs: (**F**,**L**). The thick arrows in Figure 6A–L indicate the margin of tumor and the thin arrows in Figure 6G–L indicate the intratumoral GFAP positive cells.

**Figure 7 ijms-22-05557-f007:**
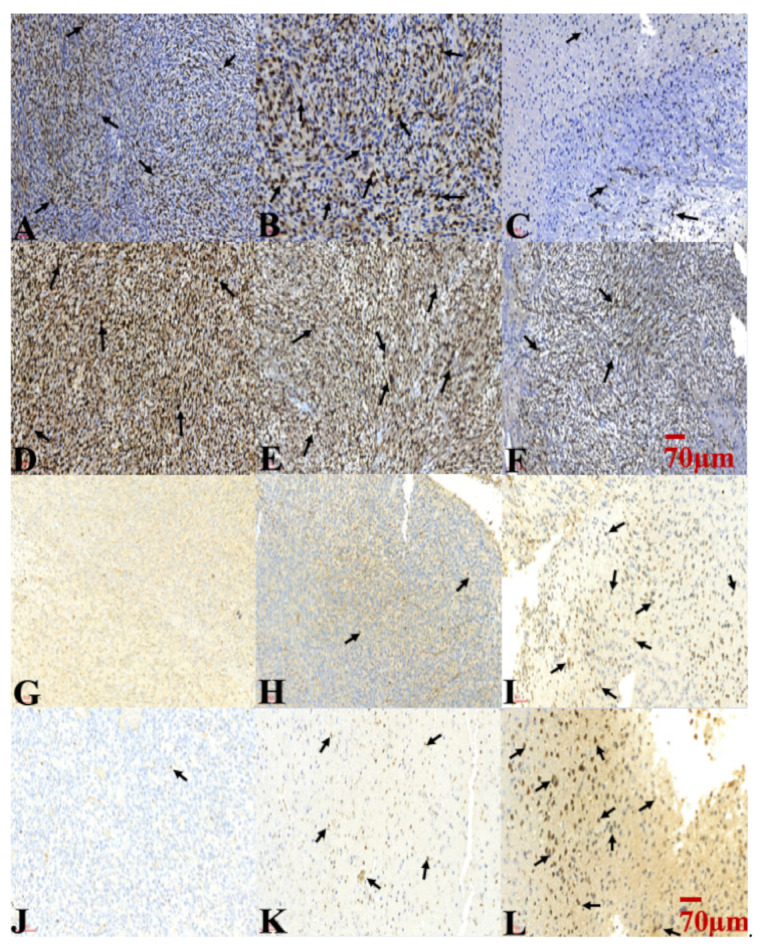
Cell proliferation and apoptosis assay were performed on U87 and U87-TR xenografted tumor regions following treatment with TMZ vehicle, TMZ-combined Gliadel, or TMZ-combined SMPs. (**A**–**F**) Ki-67 labeling was used to assess cell proliferation, and (**G**–**L**) TUNNEL assay was used to assess apoptosis. Mice bearing U87 xenografts treated with TMZ vehicle: (**A**,**G**); TMZ-combined Gliadel: (**B**,**H**); TMZ-combined SMPs: (**C**,**I**). Mice bearing U87-TR xenografts treated with TMZ vehicle: (**D**,**J**); TMZ-combined Gliadel: (**E**,**K**); TMZ-combined SMPs: (**F**,**L**). Ki-67 labeling index was performed through MIB-1 immunostaining in each subgroup. The letters in the lower-left corner of each image indicate the subgroup, followed by its Ki-67 labeling index (percentage). The arrows in panels (**A**–**F**) indicate Ki-67-positive cells. Evaluation of apoptosis of glioma cells by TUNEL assay was performed in each subgroup. The black arrows in panels (**G**–**L**) indicate TUNEL-positive apoptotic cells.

## Data Availability

Not applicable.

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
