# Peer review of "Enhanced Anti-Tumor Activity in Mice with Temozolomide-Resistant Human Glioblastoma Cell Line-Derived Xenograft Using SN-38-Incorporated Polymeric Microparticle"

_ijms, 2021, doi:10.3390/ijms22115557_

Round 1

Reviewer 1 Report

Int. J. Mol. Sci 1148830

Enhanced anti-tumor activity in mice with temozolomide-resistant human glioblastoma cell line-derived xenograft using SN-38-incorporated polymeric microparticle

Summary

The authors assess the anti-tumor activity of SN-38-loaded PLGA microparticles (SMP) in combination with TMZ in a murine xenograft model derived from both TMZ-sensitive and TMZ-resistant human glioblastoma cell lines. The authors claim that the combination therapy reduces tumor growth and extends survival of the tumor-bearing mice compared to treatment with SN-38-loaded PLGA microparticles alone.

Major Comments

  1. Line 214. The authors report an average SMP diameter of 1.26 +/- 0.79 um however the particles presented in Figure 1A are significantly larger. Please clarify this discrepancy. Additional characterization is also required including the polydispersity, zeta potential, drug loading efficiency, and drug loading capacity of the microparticles.
  2. Figure 1B. The scale of the third panel in Figure 1B appears to be significantly smaller compared to the first two panels. As such, it is difficult to assess the inflammatory response from the figures provided. Include scale bars within each panel of Figure 1B and provide representative images at similar scales. Furthermore, please describe the anatomical location where the SMPs were injected in the text (i.e., line 217). Does resolution of the inflammatory response coincide with resorption and clearance of the PLGA microparticles from the body?
  3. Figure 1C. Replot this data without a 3D perspective and include properly labeled axis titles, error bars, statistical significance, etc.
  4. Line 183. Include a more complete description of how was the Gliadel wafer (1/8th of a full wafer) inserted into the brain. Was this accomplished using a needle or was it surgically implanted through a large opening in the skull? Please also describe the how the Gliadel wafer size and corresponding BCNU dose was selected, and justify why this dose is appropriate for the present study.
  5. Section 3.2. Cytotoxicity of the SMPs should be evaluated in U87-TR cells in addition to the U87 and HA-h cells.
  6. Figure 1D. For the 72-hour timepoint, how do the authors explain the low (~50%) U87 cell viability at the lower SN-38 concentrations? This prevents accurate determination of the IC50. The resolution of this figure is poor, please include a higher resolution figure. In addition, cytotoxicity of the microparticles in a F98 experimental group is presented, but not reported in the text.
  7. The authors need to determine whether the combination therapy is additive or synergistic. As such the in vitro cytotoxicity of the mono therapies (SMPs and TMZ) as well as the combination treatment (SMPS and TMZ) should be evaluated in both U87 and U87-TR cells.
  8. The authors need to justify the use of SMPs over free drug (SN-38). As such the authors should perform in vitro cytotoxicity studies comparing the cell viability of U87 and U87-TR cells treated with SMPs and an equivalent dose of free SN-38 (based on released SN-38 drug at the various time points). The in vitro release characteristics of the SMPs must also be included in the manuscript.
  9. Significant necrosis is expected to occur in the present tumor model. TUNEL staining identifies DNA fragmentation, a characteristic of both apoptotic as well as necrotic cells. To discriminate apoptotic cells from necrotic cells, an additional method of apoptosis detection in addition to TUNEL should be performed. One such method is caspase-3 immunohistochemical labeling as it is a more direct, specific and earlier marker of apoptosis compared to TUNEL.

Minor Comments

  • Line 99. In contrast to previous statements within the introduction, the authors note that SN-38 has low toxicity on line 99. Is this a mistake? Did the authors mean to state that SN-38 SMP’s have low toxicity?
  • Please include a more complete description of the microparticle preparation method within the manuscript.

Author Response

Major Comments

  1. Line 214. The authors report an average SMP diameter of 1.26 +/- 0.79 um however the particles presented in Figure 1A are significantly larger. Please clarify this discrepancy. Additional characterization is also required including the polydispersity, zeta potential, drug loading efficiency, and drug loading capacity of the microparticles.

Response: Thank you for your rigorous comment. The Fig 1A was replaced to a more correct one. And we also measured zeta potential and polydispersity index and those were -0.84±0.1 mV and 3.657, respectively. All the data was added in revised manuscript. We developed SN-38 loaded biocompatible and biodegradable poly(lactid-co-glycolid) (PLGA) microparticles (SMPs) by electrospraying technique. The sizes of SMPs were tiny and can be mixed with solution and Local and direct injection of SMPs to the target site can provide the advantages of achieving sustained/controlled drug release, high anticancer activity and low systemic side effects. In this study, we evaluated this therapy and compared it with other treatment regimens in xenografts that are either derived from TMZ-resistant (U87TR) or TMZ-sensitive (U87)human glioblastoma cell lines. Since we did not change the loaded drug (SN-38), vehicle (PLGA) and fabrication method (electrospraying), the studies included in vitro/ in vivo, and drug loading capacity were not repeatedly performed. More data related physicochemical characterization of SMPs had described in our previous study [Pharmaceutics 2020, 12. doi:10.3390/pharmaceutics12050479].

  1. Figure 1B. The scale of the third panel in Figure 1B appears to be significantly smaller compared to the first two panels. As such, it is difficult to assess the inflammatory response from the figures provided. Include scale bars within each panel of Figure 1B and provide representative images at similar scales. Furthermore, please describe the anatomical location where the SMPs were injected in the text (i.e., line 217). Does resolution of the inflammatory response coincide with resorption and clearance of the PLGA microparticles from the body?

Response: Thank you for your comments. We hand revised the Fig 1B, and change to the same magnification and clear image. The scale bar was also added in the figure 1B. The location of SMPs injection was better described in revised manuscript “Approximately 6-mm length linear scalp incision was performed lateral to the midline. The undermined muscle and scalp fascia were dissected, and a burr hole (approximately 1.0 mm in diameter) was drilled using an electric burr. “ (Line 172-175).

“The thick arrows indicated the injected SMPs and thin arrows indicated leukocytes.” and it was addressed the Figure caption of Figure 1. The inflammatory response subsided with resorption and clearance of the PLGA microparticles from the body.

  1. Figure 1C. Replot this data without a 3D perspective and include properly labeled axis titles, error bars, statistical significance, etc.

Response: Thank you so much for your careful check. Figure 1C was revised and it displayed brain/ blood concentration ratio of SN-38 in each time-point. Since the data were presented as brain/ blood ratio, that lack of error bars and statistical significance. The Y-axis stands for brain/ blood concentration ratio and the X-axis stands for time (days).

  1. Line 183. Include a more complete description of how was the Gliadel wafer (1/8thof a full wafer) inserted into the brain. Was this accomplished using a needle or was it surgically implanted through a large opening in the skull? Please also describe the how the Gliadel wafer size and corresponding BCNU dose was selected, and justify why this dose is appropriate for the present study.

Response: Thank you so much for your careful check. Currently, the only commercially available agent for local delivery chemotherapy of brain tumor is Gliadel, which was implanted into brain cavity after removal of brain tumor. In this study, 1/8th Gliadel wafer was break into smaller pieces and infused to DMSO solution. Then the solution was introduced into brain parenchyma through a small burr hole via stereotactic techniques instead of surgical craniotomy.

The BCNU dose had described in the manuscript as “ The TMZ-combined Gliadel treatment cohort: mice in this group were given a Gliadel treatment using a stereotactic intratumoral injection of 3 µL of DMSO mixed with one-eightieth Gliadel (2.5 mg) that contains 96.25 µg of BCNU.” in line 203-206. The ratio of PLGA to SN-38 was 6:1 in fabrication of SMPs (300 mg of PLGA and 50 mg of SN-38). Therefore, we used 0.6 mg microparticles in the in vivo animal tests, corresponding to approximately 100 µg SN-38 in the SMPs group.

  1. Section 3.2. Cytotoxicity of the SMPs should be evaluated in U87-TR cells in addition to the U87 and HA-h cells.

Response: Thank you for your comments. Our U87 and U87-TR cells were kindly provided by Professor Jian-Yin Chuang. U87-TR was derived from U87 glioma cell. Cells were incubated in Dulbecco’s modified Eagle’s medium (Thermo Fisher Scientific,Waltham, MA, USA) with 10% fetal bovine serum (Thermo Fisher Scientific), 100 U/mL penicillin, and 100 g/mL streptomycin. TMZ-resistant cells were maintained in the same culture medium containing 50 μM or 100 μM TMZ (Sigma-Aldrich, St. Louis, MO, USA) as indicated. [Cancers 2020, 12(4), 981; Redox Biol. 2017 Oct;13:655-664]. Since the two cell lines only change their sensitivity to temozolomide, we did not performed cytotoxicity of the SMPs should be evaluated in U87-TR cells in addition to the U87 and HA-h cells.

  1. Figure 1D. For the 72-hour timepoint, how do the authors explain the low (~50%) U87 cell viability at the lower SN-38 concentrations? This prevents accurate determination of the IC50. The rsolution of this figure is poor, please include a higher resolution figure. In addition, cytotoxicity of the microparticles in a F98 experimental group is presented, but not reported in the text.

Response: Thank you for your comments. It is true that at low (~50%) of U87 cell viability at the lower SN-38 concentrations may not presents accurate determination of the IC50 for 72 hours. But for 48hours, we can calculate IC50 of SN-38 for U87. We think it is not important to calculate the accurate IC50 for 72 hours. Figure 1D just presents U87 cells are more sensitive to SN-38 than normal human astrocytes at two different time points. We changed a higher resolution figure for Figures 1D. F98 group is not included in this experiment.

  1. The authors need to determine whether the combination therapy is additive or synergistic. As such the in vitro cytotoxicity of the mono therapies (SMPs and TMZ) as well as the combination treatment (SMPS and TMZ) should be evaluated in both U87 and U87-TR cells.

Response: Thank you for your comments. A well management of malignant gliomas (MGs) may require a multidisciplinary approach. While the prevalent standard treatment consists primarily of surgical debulking followed by radiation therapy and possible chemotherapy. Currently, oral temozolomide is the standard treatment in MG patient. Gliadel wafers are the only interstitial chemotherapeutic agent used for the treatment of MGs and it was usually combined with oral temozolomide for treatment of MGs. We therefore compared the SMPs developed in this study with the Gliadel wafers. The mechanisms of TMZ (DNA alkylating agent) and SMP (DNA topoisomerase I inhibitor) are different. In our study, we proposed to investigate the antitumor response of SMP in TMZ-resistance to find the alternative treatment of malignant glioma.

  1. The authors need to justify the use of SMPs over free drug (SN-38). As such the authors should perform in vitro cytotoxicity studies comparing the cell viability of U87 and U87-TR cells treated with SMPs and an equivalent dose of free SN-38 (based on released SN-38 drug at the various time points). The in vitro release characteristics of the SMPs must also be included in the manuscript.

Response: Thank you for your comments. Local injection of antimitotic drugs into cerebral tumors had been investigated. Nevertheless, due to vascularity of the glioma tissue bed high enough concentrations of drug could not remain for an adequate period of time. (Science. 2005 May 27; 308(5726):1314-8). Furthermore, SN-38 is hydrophobic and is practically insoluble in most physiologically compatible and pharmaceutically acceptable solvents. Even hydrophobic molecules which can cross the BBB are frequently pumped back into the bloodstream by astrocytes closely associated with the endothelium (Nature Rev Neurosci 2006;91:41). Besides, the SN-38 is highly toxic so only few animal studies directed intravenous infusion with low-dose SN-38 to treat non-CNS tumor (Adv Drug Deliv Rev. 2011 Mar 18;63(3):184-92). One way of improving the solubility and stability of SN-38 is to formulate the drug into polymeric particles, lyposome and other vehicles (PDA J Pharm Sci Tech 2009; 63(6):512-520; Int J Nanomedicine. 2018; 13: 2789–2802). Therefore, we did not use intratumoral injection of free SN-38 (free drug) as the control group.

In this study, we developed SN-38 loaded biocompatible and biodegradable poly(lactid-co-glycolid) (PLGA) microparticles (SMPs) by electrospraying technique. The sizes of SMPs were tiny and can be mixed with solution and introduced into brain parenchyma via stereotactic techniques instead of surgical craniotomy. Local and direct injection of SMPs to the target site can provide the advantages of achieving sustained/controlled drug release, high anticancer activity and low systemic side effects. Furthermore, stereotactic injection can be repeated safely and is considered the procedure of choice in cases of multiple and/or deeply tumors, for which an open surgical approach is impossible and/or perilous, and when the tumor is located at crucial areas of the brain. The in vitro daily and cumulative release curves of SN-38 from the SMPs are presented in our previous study [Pharmaceutics 2020, 12, doi:10.3390/pharmaceutics12050479] and it was cited in the manuscript.

  1. Significant necrosis is expected to occur in the present tumor model. TUNEL staining identifies DNA fragmentation, a characteristic of both apoptotic as well as necrotic cells. To discriminate apoptotic cells from necrotic cells, an additional method of apoptosis detection in addition to TUNEL should be performed. One such method is caspase-3 immunohistochemical labeling as it is a more direct, specific and earlier marker of apoptosis compared to TUNEL.

Response: We are appreciative of the reviewer’s suggestion. This manuscript describes the antitumor activity of SN-38 embedded poly[(d,l)-lactide-co-glycolide] (PLGA) microparticles (SMPs) when used in combination with the standard GBM drug Temozolomide in a mice drug (TMZ) resistant tumor model. We focus on anti-tumor activity, increase extent survival time, decreased tumor volume, attenuated malignancy in mice bearing xenograft tumors derived from both TMZ-resistant and TMZ-sensitive human glioblastoma cell lines. Our findings demonstrate that combining SMPs with TMZ may have potential as a promising strategy for the treatment of GBM. In our further study, we will follow apoptosis guidelines (Cell Death Differ. 2009 doi:10.1038/cdd.2009.44), combine at least two distinct methods included caspase-3 immunohistochemical labeling that assess end-stage cell death.

Minor Comments

  1. Line 99. In contrast to previous statements within the introduction, the authors note that SN-38 has low toxicity on line 99. Is this a mistake? Did the authors mean to state that SN-38 SMP’s have low toxicity?

Response: Thank you for your comments. We mean the SN-38 release from SMPs has much less toxicity than that of direct (intrvenous administration) use of SN-38.

  1. Please include a more complete description of the microparticle preparation method within the manuscript.

Response: Thank you for your rigorous consideration. The preparation method of both virgin PLGA microparticles and SMPs and the process of incorporating SN-38 into the PLGA microparticles were described in our previous publication [Pharmaceutics 2020, 12, doi:10.3390/pharmaceutics12050479]. We also make brief description in section 2.5 in the revised manuscript (Line 147-159).

Reviewer 2 Report

The paper entitled “Enhanced anti-tumor activity in mice with temozolomide-resistant human glioblastoma cell line-derived xenograft using” by Tao-Chieh Yang et al. tested the anti-tumor activity of SN-38-incorporated polymeric microparticle (SMP)-based interstitial chemotherapy combined with TMZ using TMZ-resistant human glioblastoma cell line-derived xenograft models.

The paper is interesting and results quite convincing but there are some concerns on the description of the data.

Major comments

A first concern is about the number of GBM lines used in the study, only one. Increasing the number of cell lines to even two could improve the value of the paper.

It is not clear how many mice did you sort in the different harms: the mice used are 24/27, treatment harms (including vehicle) are four (vehicle, TMZ, TMZ+Gliadel, TMZ+SMP) so 6 mice/group. Why in Fig. 2 and 3 you mention 8 animals/group? Please specify!

Standard deviation of SMP treated mice in Fig. 3 are not convincing, why so narrow? From IVIS images of Fig. 4 and 5 is evident a large heterogeneity of tumor growth in SMP treated mice!

The authors assert that SMP treatment cause apoptosis but this reviewer think that with the data shown they could not achieve this conclusion. Following apoptosis guidelines (e.g. Cell Death Differ. 2009 doi:10.1038/cdd.2009.44), investigators need to combine at least two distinct methods that assess end-stage cell death. The reviewer thinks that this result could not be conclusive on the presence of apoptosis, and additional tests should be included in the paper!

The description of the presence/absence of metastases should be included in the results section, leaving in the discussion section only comments on this result.

Minor comments

Please check the type font throughout the entire manuscript there many italic or bold words that should be converted in round font, except when necessary.

Please unify the acronym of the U87-MG cell line, often is U87 and sometimes is U87-MG.

In Fig.1D, is shown a sample (F96) that is never mentioned, what is that?

Fig. 6: please specify in the legend what arrows indicate.

Line 352:  please revise the sentence; is “apoptosis” or “proliferation”? not “apoptosis proliferation”!

Author Response

Major comments

  1. A first concern is about the number of GBM lines used in the study, only one. Increasing the number of cell lines to even two could improve the value of the paper. It is not clear how many mice did you sort in the different harms: the mice used are 24/27, treatment harms (including vehicle) are four (vehicle, TMZ, TMZ+Gliadel, TMZ+SMP) so 6 mice/group. Why in Fig. 2 and 3 you mention 8 animals/group? Please specify!

Response: Thank you for your comments. In this study, we had three treatment group (Vehicle, Gliadel and SMP), all these three group was combined treated with oral temozolomide. Of the successfully established xenografted tumors,there were 28 in mice bearing U87 xenografts and 27 in mice bearing U87-TR xenografts. We utilized 24 mice from each group for survival and tumor growth rate analyses. Four mice in U87 xenografts and three mice in U87-TR xenografts were used for assessing the histopathology and morphologic characteristics of the tumor tissues (Line 185-189). The text has been revised to provide better explanation.

  1. Standard deviation of SMP treated mice in Fig. 3 are not convincing, why so narrow? From IVIS images of Fig. 4 and 5 is evident a large heterogeneity of tumor growth in SMP treated mice!

Response: Thank you for your comments. We had carefully reviewed and run our data. The standard deviation in SMP-treated group is truly narrower than the other two groups. We demonstrated some of the SMP-treated subjects of Fig. 4 and Fig. 5. The figures we provided were shown as two-dimensional pictures. The tumor burden of three-dimensional imaging showed well-controlled tumor growth of all subjected in SMP-treated group. 

  1. The authors assert that SMP treatment cause apoptosis but this reviewer think that with the data shown they could not achieve this conclusion. Following apoptosis guidelines (e.g. Cell Death Differ. 2009 doi:10.1038/cdd.2009.44), investigators need to combine at least two distinct methods that assess end-stage cell death. The reviewer thinks that this result could not be conclusive on the presence of apoptosis, and additional tests should be included in the paper!

Response: We are appreciative of the reviewer suggestion. This manuscript mainly focus on the antitumor activity of SN-38 embedded poly[(d,l)-lactide-co-glycolide] (PLGA) microparticles (SMPs) when used in combination with the standard GBM drug Temozolomide in a mice drug (TMZ) resistant tumor model. We focus on anti-tumor activity of SMPs. The results of this current study also demonstrated lower Ki-67 index, higher GFAP positive cell, and less tumor necrosis area. The Ki-67 and GFAP have been the most commonly used immunohistochemical examination for GBM, despite other immunohistochemical profiles such as EGFR, p53, and P-GBLs/ S-GBLs are good migration/metastasis/invasion marker of malignant glioma. In this study, we provided not only pathological results but also MRI images, tumor volume change and survival time to confirm the therapeutic efficacy of SMNs. We appreciate the reviewer’s comments which will help with our further study.

  1. The description of the presence/absence of metastases should be included in the results section, leaving in the discussion section only comments on this result.

Response: We gratefully appreciate for your valuable comment. We description the presence of spinal metastases in the result section. Line 305-311.

Minor comments

  1. Please check the type font throughout the entire manuscript there many italic or bold words that should be converted in round font, except when necessary.

Response: We feel sorry for the inconvenience brought to the reviewer and the mistakes and corrected in to revised manuscript.

  1. Please unify the acronym of the U87-MG cell line, often is U87 and sometimes is U87-MG.

Response: Thank you for your comments. All “U87-MG” have been revised to “U87” in the revised manuscript.

  1. In Fig.1D, is shown a sample (F96) that is never mentioned, what is that?

Response: We apologized the mistakes and the Fig. 1D was changed to high resolution one and the data of F98 group was removed from Fig. 1D.

  1. 6: please specify in the legend what arrows indicate.

Response: Thank you for your comments. The thick arrows in Fig A-L indicated the margin of tumor and the thin arrows in Fig G-L indicated the intratumoral GFAP positive cells. The sentences had been added into the figure caption of Figure 6.

  1. Line 352:  please revise the sentence; is “apoptosis” or “proliferation”? not “apoptosis proliferation”!

Response: Thank you for your comments. In revised manuscript, the “proliferation: was deleted. (Line 383)

Reviewer 3 Report

This manuscript describes the antitumor activity of SN-38 embedded poly[(d,l)-lactide-

co-glycolide] (PLGA) microparticles (SMPs) when used in combination with the standard GBM drug Temozolomide in a mice drug (TMZ) resistant tumor model. The study was a follow up of a previous study that presented the delivery efficiency and drug release capability of the SMPs in brain tissues of rats. The study used mainly the live in vivo imaging, Immunocytochemistry imaging and survival curves to claim the therapeutic potential of SMPs in combination with TMZ for the treatment of GBM in mice drug resistant tumor model. Although I feel that a mechanistic approach to fully support the potency of this combinatorial approach is lacking this study is promising and will be of interest to the researchers of the dreaded brain cancer, GBM. I suggest to address the following queries to improve the quality of the manuscript for publication.

  1. Include western blot and/or RT-PCR data of MGMT expression levels in U87 and U87-R cell lines to indicate the mechanistic impact of SMPs in TMZ resistance.
  2. Figure 1B: It is not clear what the arrows indicate. The H&E images appears to be of different magnification, so indicate the scale bars in each image.
  3. Lane 184: The SMP was mixed with 3 ul DMSO for injection. SN-38 is highly soluble in DMSO, so will that result in the premature SN-38 release from the PLGA microparticles? Why don’t the authors use HBSS or similar compatible physiological solution to introduce SMPs to the brain?
  4. Figure A,C,D &F have poor resolution.
  5. Give details/methods of in vivo drug release study.

Author Response

  1. Include western blot and/or RT-PCR data of MGMT expression levels in U87 and U87-R cell lines to indicate the mechanistic impact of SMPs in TMZ resistance.

Response: Thank you for your comments. The cell lines, including U87 and U87TR, were obtained from Professor Jian-Yin Chuang’s lab. The antitumor response and their expression of MGMT can be found in the literature and figures [Cancers 2020, 12(4), 981; Redox Biol. 2017 Oct;13:655-664]. The literatures were cited in this paper. The western blot revealed both the U87 and U87-TR were MGMT negative. (Figure 1E in Redox Biol. 2017 Oct;13:655-664.) The two references [5] [32] have been added into the reference list and cited in the text.

  1. Figure 1B: It is not clear what the arrows indicate. The H&E images appears to be of different magnification, so indicate the scale bars in each image.

Response: Thank you for your comments. We hand revised the Fig 1, and change to the same magnification and high resolution images. The thick arrows indicated the injected SMPs and thin arrows indicated leukocytes, and it was addressed the Figure caption of Figure 1.

  1. Lane 184: The SMP was mixed with 3 ul DMSO for injection. SN-38 is highly soluble in DMSO, so will that result in the premature SN-38 release from the PLGA microparticles? Why don’t the authors use HBSS or similar compatible physiological solution to introduce SMPs to the brain?

Response: Thank you for your comments. SN-38 is poorly soluble in aqueous solutions, and it is practically insoluble in most physiologically compatible and pharmaceutically acceptable solvents. We tried to use normal saline and PBS solution to mix with SMPs firstly, but the microparticles could not be mixed well with these two solutions. Small amount (10 µL) of DMSO was thus employed as an assisting fluid to promote the injection of SN-38 particles into the brains of the animals. The SMPs were not highly soluble in DMSO, indeed the SMPs were infused in DMSO solution and were stereotactically injected into brain via Neurosyringe. For future clinical trial of SMPs, other assisting fluids may be used.

  1. Figure A,C,D &F have poor resolution.

Response: Thank you for your comments. The Figure 7 was revised and the resolution of Fig. 7 was improved obviously.

  1. Give details/methods of in vivo drug release study.

Response: Thank you for your comments. The methods of in vivo drug release was added as section 2.5 in revised manuscript. (Line 147-159)

Round 2

Reviewer 1 Report

Major Comments

A number of comments remain unaddressed or inadequately addressed including:

  • Line 214. Additional characterization is also required including the drug loading efficiency and drug loading capacity of the microparticles. This data is critical to determine the dose of the microparticles. Currently the authors are assuming 100% drug loading efficiency which is not correct and will impact interpretation of all subsequent experiments: “The BCNU dose had described in the manuscript as “ The TMZ-combined Gliadel treatment cohort: mice in this group were given a Gliadel treatment using a stereotactic intratumoral injection of 3 µL of DMSO mixed with one-eightieth Gliadel (2.5 mg) that contains 96.25 µg of BCNU.” in line 203-206. The ratio of PLGA to SN-38 was 6:1 in fabrication of SMPs (300 mg of PLGA and 50 mg of SN-38). Therefore, we used 0.6 mg microparticles in the in vivo animal tests, corresponding to approximately 100 µg SN-38 in the SMPs group.” Critical data such as this should be included in the present manuscript rather than previous publications.

  • Section 3.2. Cytotoxicity of the SMPs should be evaluated in U87-TR cells in addition to the U87 and HA-h cells.

  • The resolution of Figure 1D remains quite low.

  • Significant necrosis is expected to occur in the present tumor model. TUNEL staining identifies DNA fragmentation, a characteristic of both apoptotic as well as necrotic cells. To discriminate apoptotic cells from necrotic cells, an additional method of apoptosis detection in addition to TUNEL should be performed. One such method is caspase-3 immunohistochemical labeling as it is a more direct, specific and earlier marker of apoptosis compared to TUNEL. As the authors mentioned, apoptosis must be assessed with more than one method. Please include this data in the present manuscript.

Author Response

  • Line 214. Additional characterization is also required including the drug loading efficiency and drug loading capacity of the microparticles. This data is critical to determine the dose of the microparticles. Currently the authors are assuming 100% drug loading efficiency which is not correct and will impact interpretation of all subsequent experiments: “The BCNU dose had described in the manuscript as “ The TMZ-combined Gliadel treatment cohort: mice in this group were given a Gliadel treatment using astereotactic intratumoral injection of 3 µL of DMSO mixed with one-eightieth Gliadel (2.5 mg) that contains 96.25 µg of BCNU.” in line 203-206. The ratio of PLGA to SN-38 was 6:1 in fabrication of SMPs (300 mg of PLGA and 50 mg of SN-38). Therefore, we used 0.6 mg microparticles in the in vivo animal tests, corresponding to approximately 100 µg SN-38 in the SMPs group.” Critical data such as this should be included in the present manuscript rather than previous publications.

Response: Thank you for your comments. The main advantageous features over conventional methods are the possibility to produce particles without the use of surfactants, at ambient temperature and pressure and with maximum encapsulation efficiency due to the absence of an external medium that allows the migration and/or dissolution of water-soluble cargos. (J Biomed Nanotechnol. 2014 Sep;10(9):2200-17. doi: 10.1166/jbn.2014.1887.)

  • Section 3.2. Cytotoxicity of the SMPs should be evaluated in U87-TR cells in addition to the U87 and HA-h cells.

Response: Thank you for your comments. The MTT test was performed in U87-TR cell line and the data was added in revised manuscript. The figure 1D was revised.

  • The resolution of Figure 1D remains quite low.

 Response: Thank you for your comments. The figure 1D was replaced to more clear one.

  • Significant necrosis is expected to occur in the present tumor model. TUNEL staining identifies DNA fragmentation, a characteristic of both apoptotic as well as necrotic cells. To discriminate apoptotic cells from necrotic cells, an additional method of apoptosis detection in addition to TUNEL should be performed. One such method is caspase-3 immunohistochemical labeling as it is a more direct, specific and earlier marker of apoptosis compared to TUNEL. As the authors mentioned, apoptosis must be assessed with more than one method. Please include this data in the present manuscript.

Response: We are appreciative of the reviewer suggestion. The present manuscript mainly focus on the antitumor activity of SN-38 embedded poly[(d,l)-lactide-co-glycolide] (PLGA) microparticles (SMPs) when used in combination with the standard GBM drug Temozolomide in a mice drug (TMZ) resistant tumor model. We focus on anti-tumor activity of SMPs. The results of this current study also demonstrated lower Ki-67 index, higher GFAP positive cell, and less tumor necrosis area. The Ki-67 and GFAP have been the most commonly used immunohistochemical examination for GBM, despite other immunohistochemical profiles such as EGFR, p53, and P-GBLs/ S-GBLs are good migration/metastasis/invasion marker of malignant glioma. In this study, we provided not only pathological results but also MRI images, tumor volume change and survival time to confirm the therapeutic efficacy of SMNs. We appreciate the reviewer’s comments which will help with our further study.

Reviewer 2 Report

Response to comments are satisfactory except for comment 2 and comment 3.

Comment 2: still not convinced the standard deviation reported in Fig. 3 left and right panels, especially looking at images in Fig. 4 day 29 and day 40, and in Fig. 5 day 22. Probably the authors must use other statistic tools. In addition in Fig. 3 right panel are not reported data concerning SMP at day 29 and day 40 that instead are shown as images in Fig. 5. Please add them in Fig. 3!

Comment 3: the analysis of apoptosis is not conclusive and not acceptable in such way. If the authors do not intend to further investigate apoptosis please remove the data from the manuscript since it is not relevant. 

Author Response

Comment 2: still not convinced the standard deviation reported in Fig. 3 left and right panels, especially looking at images in Fig. 4 day 29 and day 40, and in Fig. 5 day 22. Probably the authors must use other statistic tools. In addition in Fig. 3 right panel are not reported data concerning SMP at day 29 and day 40 that instead are shown as images in Fig. 5. Please add them in Fig. 3!

Response: Thank you for your comments. We checked our data, there was no U87-TR bearing mouse survival at 40 day in the vehicle group and only one mouse survival on day 40 in Gliadel group. The figure 3 and figure 5 were revised and we apologized the mistakes. The standard deviation in SMP-treated group is much narrower in SMP group than the other two groups. We demonstrated some of the SMP-treated subjects of Fig. 4 and Fig. 5. The figures we provided were shown as two-dimensional pictures. The tumor burden of three-dimensional imaging showed well-controlled tumor growth of all subjected in SMP-treated group. The figure 5 was be revised to a more suitable one.

Comment 3: the analysis of apoptosis is not conclusive and not acceptable in such way. If the authors do not intend to further investigate apoptosis please remove the data from the manuscript since it is not relevant. 

Response: We are appreciative of the reviewer suggestion. The present manuscript mainly focus on the antitumor efficacy of SN-38 embedded poly[(d,l)-lactide-co-glycolide] (PLGA) microparticles (SMPs) when used in combination with the standard GBM drug Temozolomide in a mice drug (TMZ) resistant tumor model. We focus on anti-tumor activity of SMPs. The results of this current study also demonstrated lower Ki-67 index, higher GFAP positive cell, and less tumor necrosis area. The Ki-67 and GFAP have been the most commonly used immunohistochemical examination for GBM, despite other immunohistochemical profiles such as EGFR, p53, and P-GBLs/ S-GBLs are good migration/metastasis/invasion marker of malignant glioma. In this study, we provided not only pathological results but also MRI images, tumor volume change and survival time to confirm the therapeutic efficacy of SMPs. We appreciate the reviewer’s comments which will help with our further study.

Round 3

Reviewer 1 Report

Minor improvements in figure quality are required (e.g., Resolution of Figure 1D, scaling of color bars in Figures 4 and 5)